# Effect of Exposure to Particulate Matter on the Ocular Surface in an Experimental Allergic Eye Disease Mouse Model

**DOI:** 10.3390/bioengineering11050498

**Published:** 2024-05-16

**Authors:** Basanta Bhujel, Seheon Oh, Woojune Hur, Seorin Lee, Ho Seok Chung, Hun Lee, Jin Hyoung Park, Jae Yong Kim

**Affiliations:** 1Department of Ophthalmology, University of Ulsan College of Medicine, Asan Medical Center, Seoul 05505, Republic of Korea; basanta@amc.seoul.kr (B.B.); sh.oh@amc.seoul.kr (S.O.); dnwnsgj@amc.seoul.kr (W.H.); ra02582@amc.seoul.kr (S.L.); chunghs@amc.seoul.kr (H.S.C.); yhun777@amc.seoul.kr (H.L.); 2Department of Medical Science, University of Ulsan Graduate School, Seoul 05505, Republic of Korea; 3MS Eye Clinic, Seongnam 13640, Republic of Korea; drpark99@naver.com

**Keywords:** particulate matter (PM), ovalbumin (OVA), allergic eye disease (AED), allergic ocular model

## Abstract

In response to the escalating concern over the effect of environmental factors on ocular health, this study aimed to investigate the impact of air pollution-associated particulate matter (PM) on ocular allergy and inflammation. C57BL/6 mice were sensitized with ovalbumin (OVA) topically and aluminum hydroxide via intraperitoneal injection. Two weeks later, the mice were challenged with OVA and exposed to PM. Three groups—naive, OVA, and OVA-sensitized with PM exposure (OVA + PM) groups—were induced to an Allergic Eye disease (AED) model. Parameters including clinical signs, histological changes, inflammatory cell infiltration, serum OVA-specific immunoglobulins E (IgE) levels, mast cells degranulation, cellular apoptosis and T-cell cytokines were studied. The results demonstrate that exposure with PM significantly exacerbates ocular allergy, evidenced by increased eye-lid edema, mast cell degranulation, inflammatory cytokines (IL-4, IL-5 and TNF-α), cell proliferation (Ki67), and serum IgE, polymorphonuclear leukocytes (PMN), and apoptosis and reduced goblet cells. These findings elucidate the detrimental impact of PM exposure on exacerbating the severity of AED. Noticeably, diminished goblet cells highlight disruptions in ocular surface integrity, while increased PMN infiltration with an elevated production of IgE signifies a systemic allergic response with inflammation. In conclusion, this study not only scientifically substantiates the association between air pollution, specifically PM, and ocular health, but also underscores the urgency for further exploration and targeted interventions to mitigate the detrimental effects of environmental pollutants on ocular surfaces.

## 1. Introduction

Air pollution has severe adverse effects on health that have become more pronounced in recent years with increased morbidity and mortality worldwide. With continued urbanization, air pollution-related health problems are expected to worsen with time [1]. Particulate matter (PM) is one of the major components of air pollution and is used to indicate its severity. Ambient levels of PM aggravate existing asthma by inducing oxidative stress and allergic inflammation [2]. The eye is one of the vulnerable organs to environmental risk because it is constantly exposed to the external environment. Harmful effects of PM on the eye, such as eye irritation, blepharitis, dry eye diseases (DED), and allergic conjunctivitis (AC), are well reported in prior studies [3,4]. Recently, increasing evidence has suggested that PM is a key component exacerbating allergic eye disease (AED) [5]. AED is an important public health threat and constitutes a significant economic burden to the general public globally [5,6]. It refers to a range of specific allergic inflammatory conditions impacting the conjunctiva, eyelid, and, in severe instances, the cornea. AED involves immunoglobulins E (IgE)-mediated mast cell activation within conjunctival tissue, resulting in the release of preformed mediators like histamine and proteases. This initial activation also leads to the synthesis of lipid-derived mediators and cytokines, initiating a complex cascade of cellular and molecular responses. These events culminate in the substantial migration and infiltration of inflammatory cells to the ocular surface [7]. The major clinical signs of AED consist of chemosis, tearing, conjunctival hyperemia, and eyelid edema. In fact, the term AED includes distinct clinical conditions, mild seasonal and perennial AC to more serious and chronic conditions like atopic and vernal keratoconjunctivitis (AKC and VKC, respectively) [5,8]. Notably, exposure to ovalbumin (OVA), a prevalent allergen, can intensify AED by inciting specific immune reactions [9]. This mirrors the intricate nature of AED, where various environmental allergens, including PM, contribute to its complexity and clinical expression [10].

Prior investigations have extensively examined the impact of PM on allergic diseases induced by OVA, particularly focusing on respiratory allergic diseases. PM influences cytokine production, enhances the IgE response, modulates the immunoglobulin isotype switching and exacerbates allergic and inflammatory lung diseases in an OVA induced allergic mouse model [11,12,13,14]. In the pulmonary allergic mice model, PM exacerbated the allergic immune response by acting as an immune adjuvant and thereby increased eosinophil migration, Th2 cytokines, IgE expression, and mucosubstance production [15]. Similarly, increment in the interlukin-5 (IL-5) levels, tumor necrosis factor (TNF-α), heme oxygenase-1 (HO-1) expression, and cellular inflammations were observed in the PM-exposed mice, thereby promoting inflammation and allergic responses in the mouse model [16,17]. In an OVA-induced allergic rhinitis mouse model, exposure to PM intensifies oxidative stress and inflammatory responses by modulating the Nuclear Factor Kappa B (NF-κB) signaling pathway [18].

In a mouse model of allergic inflammation on an ocular surface, OVA-sensitized with PM exposure has been shown to exacerbate inflammation via promoting Th2 responses and increasing the infiltration of inflammatory cells such as interleukin-1 beta (IL-1β), interleukin-6 (IL-6), interleukin-17 (IL-17), and TNF-α in the conjunctiva upon challenge with the allergen [9]. In the context of OVA-induced IgE-mediated ocular allergic reactions triggered by exposure to PM, there is a subsequent infiltration primarily dominated by mast cells and eosinophils. This cellular influx leads to the generation of diverse inflammatory cytokines [10,19,20,21]. The scenario induced by PM or related airborne particulates included autophagy, DNA damage, and cell senescence in corneal epithelial cells, an increased production of pro-inflammatory cytokines and mucin, and tear osmolarity changes [9]. The immune-pathogenic mechanisms underlying AED entail responses mediated by either IgE or an allergenic cluster of differentiation 4 (CD4^+^) T helper Th2 cells and their cytokines like interlukins-4, 5 and 13 (IL-4, IL-5, IL-13) [22,23,24]. However, very few studies have evaluated the relationship between the PM and AED pathogenesis while most of the studies have investigated the influence of diesel exhaust particles and titanium dioxide on eye diseases, which did not accurately reflect the characteristics of PM in the eyes [25,26].

Considering the prevailing conditions and the potential influence of PM exposure on the exacerbation of respiratory allergic diseases, our hypothesis posits that the co-exposure to PM along with OVA may similarly amplify the symptoms associated with AED, as it mimics allergic reactions in human ocular surfaces. To validate our hypothesis, in this study, we established an OVA-induced with PM exposed AED mouse model with the standard reference material (SRM 2786, mean diameter <4 μm) of PM powder containing organic/inorganic constituents. The particle-size characteristics of this PM powder align with those of atmospheric particulate materials and comparable matrices, differing somewhat from the finer PM utilized in the majority of prior studies [27,28]. This study aimed to explore the possible adjuvant activity of ambient fine PM in triggering immune responses and inflammation and thereby contributes to allergic sensitization in a mouse model of AED (Figure 1A) simulating a human ocular surface, in conjunction with OVA.

## 2. Materials and Methods

### 2.1. Animals

This study was conducted in strict accordance with and adherence to the relevant national and international guidelines regarding animal handling as mandated by the Institutional Animal Care and Use Committee (IACUC) of the Ulsan University. All animal experiments were performed according to the ARVO Announcement on Animal Use in Ophthalmology and Visual Studies and were reviewed and authorized by the Animal Care and Use Committee of the Life Sciences Institute of Asan Medical Center. The committee complies with the Laboratory Animal Resources Guidelines.

Thirty female C57BL/6 mice (Orient Bio, Inc., Seongnam, Republic of Korea), aged 8 weeks, underwent a 1 week acclimation period. They were housed under a 12 h light/dark schedule with access to autoclaved food and water ad libitum. The mice were cared for in accordance with ethical standards to minimize any potential suffering and were housed in a specific pathogen-free facility.

### 2.2. Allergic Sensitization and PM Exposure in Mouse

The AED induction was performed using a previously described model [7]. In brief, to generate the AED mouse model, all mice were sensitized intraperitoneally with 10 μg of ovalbumin (OVA, Grade V; Sigma-Aldrich, Inc., St. Louis, MO, USA), 1 mg of aluminum hydroxide (ALUM, Pierce, Rockford, IL, USA), and 300 ng of pertussis toxin (Sigma-Aldrich, Inc., St. Louis, MO, USA) on day 1 for one time. The mice were randomly divided into three groups: (1) naive, (2) OVA, and (3) OVA + PM (*n* = 10/each group). The naive group consisted of mice that only underwent sensitization without further challenge. Following the sensitization, a 14-day rest period was allowed.

Mice in the OVA and OVA + PM groups were challenged with 250 µg/5 µL OVA eye drops topically once a day. Additionally, at the same time, mice in the OVA + PM group were further exposed to 3 mg/mL of PM (SRM 2786; Sigma-Aldrich, Inc.) three times a day from days 15 to 21 (Figure 1B).

After the successful induction of the AED mouse model, all the mice were euthanized at day 22 via CO_2_ asphyxiation and the left whole eyes were harvested for histological, immunohistological and protein analyses.

### 2.3. Clinical Scoring

The ocular manifestations of AED were assessed utilizing a slit lamp microscope. Blind methods were used to score different degrees of conjunctival hyperemia, conjunctival chemosis, eyelid edema, and secretions by clinical ophthalmologists. The clinical scores range from 0 to 3 points for each parameter [29]. Finally, the total score was calculated by summing the scores of the four indicators, ranging from a minimum of 0 points to a maximum of 12 points [30,31].

Briefly, in the context of conjunctival hyperemia, a score of 0 signifies the absence of redness, while scores 1 to 3 correspond to increasing levels of severity, ranging from bright red blood vessels to dark red vessels and ultimately to a diffuse purple appearance. Conjunctival chemosis is evaluated on a similar scale, where higher scores represent greater degrees of swelling, from slight edema to deep edema and, potentially, to epicontinental edema.

The scoring system for eyelid edema ranges from 0 to 3, with scores corresponding to the absence of edema, minor eyelid valgus, semi-closed eyelids, and eyelids being more than half closed, respectively. Lastly, the secretion symptom is assessed based on the presence and quantity of discharge, progressing from no observable secretion to little secretion, a dampness of eyelids and eyelashes, and finally, wetness throughout the week.

### 2.4. Serum Preparation and Enzyme-Linked Immunosorbent Assay (ELISA) for OVA-Specific IgE in Serum

After the successful induction of the AED mouse model, blood was collected using cardiac puncture from the animals into centrifuge tubes and stored at low temperatures [9]. The blood was centrifuged at 3000× *g*/min for 15 min at 4 °C; the sera were isolated after coagulation with centrifugation and then analyzed for OVA-specific IgE using an ELISA kit according to the manufacturer’s instructions (LEGEND MAX^TM^, Biolegend, San Diego, CA, USA).

### 2.5. Hematoxylin and Eosin Staining (H&E)

The animals were euthanized using CO_2_ asphyxiation and the left whole eyes were harvested for histological analysis to demonstrate PMN infiltration and hemagiectasis in the mouse conjunctiva. The tissues were then processed for paraffin embedding and sectioning (5 µm) using microtome (Leica, Wetzlar, Germany). The paraffin sections underwent dewaxing using xylene I and xylene II, followed by dehydration using anhydrous ethanol I, anhydrous ethanol II, and 75% alcohol. Subsequently, the sections were stained with hematoxylin solution (Abcam, Cambridge, UK) for 3–5 min, rinsed with tap water, underwent liquid differentiation, returned to blue, and rinsed again. Dehydration was performed using 85% and 95% gradient alcohol, followed by staining with eosin dye solution (Abcam, Cambridge, UK) for 5 min. After staining, the sections were cleared with xylene and sealed with neutral gel for microscopic examination. The PMN cell number and hemangiectasis in the conjunctiva were measured using ImageJ software (https://imagej.nih.gov/ij/ accessed on 4 January 2024 developed by Wayne Rasband, National Institutes of Health, Bethesda, MD, USA).

### 2.6. Toluidine Blue Staining

The animals were euthanized using CO_2_ asphyxiation and the left whole eyes were harvested for studying the mast cell degranulation in the mouse conjunctiva. Tissues were then processed for paraffin embedding and sectioning (5 µm) using microtome (Leica, Wetzlar, Germany). Paraffin sections were dewaxed and stained with 1% toluidine blue solution (Daejeon, Republic of Korea) for 5–8 min. The stain solution, prepared through dissolving 1 g of toluidine blue in 100 mL of distilled water with 2 mL of mordant, was incubated at 60 °C for 2 h before use. After staining, sections were washed, differentiated in 95% ethanol, dehydrated, and sealed. The microscopic examination showed purple or blue granules [31]. The degranulated mast cell numbers in the conjunctiva were measured using ImageJ software (version 1.62f; https://imagej.nih.gov/ij/; accessed on 4 January 2024 developed by Wayne Rasband, National Institutes of Health, Bethesda, MD, USA).

### 2.7. Immunofluorescence and Immunohistochemistry Staining

The animals were euthanized using CO_2_ asphyxiation and the left whole eyes were harvested for immunohistochemical analysis in the mouse conjunctiva and cornea. The tissues were then processed for paraffin embedding and sectioning (5 µm) using microtome (Leica, Wetzlar, Germany). For the immuno-staining, tissue sections underwent dewaxing, rehydration, and antigen retrieval. Subsequently, the sections were blocked with a 0.1% BSA block solution supplemented with normal goat serum for 30 min at room temperature. After that, they were then incubated at 4 °C overnight with antibodies against IL-4 (1:200, Catalog No.sc-53084; Santa Cruz Biotechnology, Dallas, TX, USA), IL-5 (1:100, Catalog No. SC 398334; Santa Cruz Biotechnology, Dallas, TX, USA) and TNF-α (1:200, Catalog No. 3707; Cell Signaling Technology, Inc., Danvers, MA, USA), Ki67 (1:200, Catalog No. ab 15580; Abcam, Inc., Cambridge, MA, USA) and Substance P (1:50, Catalog No. ab14184; Abcam, Inc., Cambridge, MA, USA). The sections were washed with PBS and incubated with Alexa Fluor 488 (1:400, Catalog No. A11008; Invitrogen, Inc., Carlsbad, CA, USA), Alexa Fluor 555 (1:400, Catalog No. A21424; Invitrogen, Inc. Carlsbad, CA, USA), Alexa Fluor 568 (1:400, Catalog No. A10042; Invitrogen, Waltham, MA, USA) and Alexa Fluor 488 (1:400, Catalog No. A11029; Invitrogen, Waltham, MA, USA) for 1 h in the dark at room temperature. Additionally, the sections underwent counterstaining with DAPI for 10 min, followed by mounting. After the immunostaining of the tissue slides, 5 samples were randomly selected from each group, and 4 sections were selected from each sample. Ultimately, images were acquired under a confocal microscope (Carl Zeiss, Inc., Jena, Germany) or an optical microscope (Olympus, Inc., Tokyo, Japan). The images were analyzed using ImageJ software (version 1.62f; available by ftp at http://rsb.info.nih.gov/nih-image; accessed on 4 January 2024; developed by Wayne Rasband, National Institutes of Health, Bethesda, MD, USA).

### 2.8. Periodic Acid-Schiff (PAS) Staining

The animals were euthanized using CO_2_ asphyxiation and the left whole eyes were harvested for PAS staining to study the presence and distribution of goblet cells within the mouse conjunctiva. The tissues were then processed for paraffin embedding and sectioning (5 µm) using microtome (Leica, Wetzlar, Germany) and subjected to a PAS stain [4]. Briefly, the slides were deparaffinized and hydrated, followed by immersion in 1% PAS (Sigma-Aldrich, Burlington, MA, USA). Then, they were washed with tap water and immersed in Schiff’s Reagent (Sigma-Aldrich, MA, USA). Afterward, they were counterstained with Harris Hematoxylin solution rinse before dehydrating and clearing. The microscopic examination and image acquisition revealed magenta/pink-colored flask-shaped granules. To count goblet cells, the average number of PAS-stained cells on the four different sections from each eye was calculated under a microscope by two blinded observers. The goblet cell numbers in the conjunctiva were measured using ImageJ software (https://imagej.nih.gov/ij/ accessed on 4 January 2024; developed by Wayne Rasband, National Institutes of Health, Bethesda, MD, USA).

### 2.9. TUNEL Assay

The animals were euthanized using CO_2_ asphyxiation and the left whole eyes were harvested for Terminal deoxynucleotidyl transferase (TdT) deoxyuridine triphosphate (dUTP) nick-end labeling (TUNEL) assay to study the presence of cellular apoptosis within conjunctival and corneal tissue. Tissues were then processed for paraffin embedding and sectioning (5 µm) using microtome (Leica, Wetzlar, Germany). In brief, the conjunctival and corneal sections were oven-baked at 60 °C for 30 min, then underwent dewaxing with xylene and dehydration with ethanol prior to TUNEL (684817910; Roche, Inc., Mannheim, Germany) labeling. After the nucleus was stained with DAPI, images were taken under a confocal microscope (Carl Zeiss, Inc.) and the number of cells was counted. The positive cell number was analyzed using ImageJ software (version 1.62f; available by ftp at http://rsb.info.nih.gov/nih-image; accessed on 4 January 2024; developed by Wayne Rasband, National Institutes of Health, Bethesda, MD, USA).

### 2.10. Western Blotting

The animals were euthanized using CO_2_ asphyxiation and the mouse conjunctival tissues were harvested. The conjunctiva sample underwent cold PBS washing followed by incubation in lysis buffer containing phosphatase and protease inhibitors. Tissue lysates were sonicated and homogenized, then centrifuged to obtain supernatants for protein quantification. Electrophoresis on SDS gels, protein transfer to PVDF membranes, and blocking with BSA in TBST buffer were conducted. After that, membranes were incubated overnight at 4 °C with specific rabbit polyclonal antibodies against phosphorylated NF-kB (pNF-kB; 1:1000, Catalog No. 3033S; Cell Signaling Technology, Inc.) and GAPDH (1:10,000, Catalog No. 2118S; Cell Signaling Technology, Inc.). The membranes were washed and incubated with an HRP-linked secondary antibody, and the bands were visualized with a chemiluminescence system (WBKLS0100; MilliporeSigma, Inc., Burlington, MA, USA). The protein expression levels were normalized to GAPDH in the same samples.

### 2.11. Statistical Analysis

The statistical analysis was performed using GraphPad Prism (version 5.01, GraphPad Software, Boston, MA, USA) for data analysis and ImageJ software (Version 1.62f, https://imagej.nih.gov/ij/; accessed on 4 January 2024; developed by Wayne Rasband, National Institutes of Health, Bethesda, MD, USA for data quantification). The data are presented as the mean ± standard deviation (SD) and a significant difference using one-way ANOVA followed by the Tukey test. A Bartlett’s test assessed the effects of multiple treatments in in vivo experiments, with *p*-values < 0.05 considered statistically significant. 

## 3. Results

### 3.1. Effect of PM Exposure on Ocular Surface and Clinical Scores in an AED Mouse Model 

A mouse model for AED was successfully induced by sensitizing the mice to OVA and exposing PM. Clinical and histopathological scores were consistent with the successful establishment of the model. The animals’ ocular surface was observed for the presence of hyperemia, chemosis, edema, mucoid discharge, redness, and itching. Several studies showed an increase in the eyelid edema, epithelial lesions, tearing, scratching behaviors and redness in the ocular surface in the PM mediated ocular allergy [3,27,32].

From our findings, it was noted that the OVA + PM exposed group showed high conjunctival redness, hyperemia, and edema followed by the OVA challenged group. Furthermore, eye-lid edema was notably observed in this group. Similarly, heightened mucoid discharge with an increment in the cloudiness was observed in the OVA + PM exposed group and OVA group. Conversely, in the naive group, no such detrimental features were observed in the mouse ocular surface, underscoring the absence of allergic manifestations (Figure 1C).

Furthermore, clinical scores were utilized to assess the severity and progression of the allergic responses in the mouse ocular surface. Our results revealed significantly lower scores in the naive group compared to the OVA + PM exposed group. Additionally, the scoring pattern for the OVA group closely resembled that of the OVA + PM exposed group (Table 1).

### 3.2. Effect of PM Exposure on Goblet Cells Numbers in Conjunctiva in an AED Mouse Model

PAS staining was employed on conjunctival tissue to quantify the presence and distribution of goblet cell numbers, crucial contributors to mucous tear production in our AED mouse model. Goblet cells, identified by mucin-filled vesicles, were enumerated in PAS-stained conjunctival sections. The reduced goblet cell number, coupled with compromised mucosal integrity and mucin production, suggests that PM might serve as a potential contributing factor to ocular allergy, as indicated by several studies [3,33].

Notably, our study revealed a significant reduction in goblet cell count in both the OVA + PM and OVA groups. On the other hand, goblet cell count remained relatively stable in the naive group (Figure 2A,D).

### 3.3. Effect of PM Exposure on Polymorphonuclear Leukocytes Cell Infiltration in Conjunctiva in an AED Mouse Model

H&E staining was performed to observe the PMN cell infiltration in conjunctiva in our AED mouse model. PMNs are efficient phagocytes and act as crucial effector cells through the release of granule products and cytokines. They are the predominant cells in the ocular allergic inflammations, as well as T cell-mediated delayed hypersensitivity reactions in the conjunctiva [34,35]. Previous studies showed that exposure to the PM enhanced the PMN levels in the animal model [36,37,38].

From our study, in the OVA and OVA + PM exposed groups, the PMN cell infiltration was significantly increased in the mouse conjunctiva. However, a minimal PMN cell infiltration was observed in the naive group (Figure 2B,E).

### 3.4. Effect of PM Exposure on Hemagiectasis in Conjunctiva in an AED Mouse Model

Similarly, the hemagiectasis was determined using H&E staining in the conjunctival tissue in our AED mouse model. Prior research demonstrated that exposure to PM in animal models can lead to vascular dilation, which may suggest an acute inflammatory response. This underscores the connection between the inflammation and allergic reactions on the ocular surface [39,40]. In our findings, it was revealed that hemagiectasis significantly increased in the OVA + PM exposed group compared to the naive group. Subsequently, a similar increase was observed in the OVA group relative to the naive group. (Figure 2C,F).

### 3.5. Effect of PM Exposure on IgE Serum Concentration in an AED Mouse Model

Subsequently, to compare the allergic response among the groups, the ELISA for the OVA-specific IgE in the serum collected from the mouse was performed. Several studies demonstrated that IgE played a major role in ocular allergic diseases [9,33,41]. From our study, compared to the OVA and OVA + PM exposed group, the OVA specific IgE level in the naive group was relatively lower. This showed that the OVA-mediated IgE allergic responses were predominantly raised in the OVA + PM exposed group with allergic symptoms (Figure 1D).

### 3.6. Effect of PM Exposure on Mast Cell Aggregation in Conjunctiva in an AED Mouse Model

The mast cell degranulation was studied with toluidine blue staining in naive, OVA and OVA + PM-exposed groups. Conjunctival mast cells have been shown to be a source of several cytokines such as IL-4 and TNF-α and play a central role in the development of ocular allergic reactions [42,43].

In our study, a significant number of mast cell degranulation was observed in OVA + PM exposed and OVA groups. Interestingly, in the naive group, the mast cell degranulation was relatively lower than in the other groups (Figure 3A,C). Similar to our study, the excessive mast cell degranulation was also observed in the mouse model of AC induced by PM [33].

### 3.7. Effect of PM Exposure on Substance P in Conjunctiva in an AED Mouse Model

The immunohistochemistry for Substance P was conducted on mouse conjunctival tissue, revealing a yellow/tan coloration under optical microscopy. As in previous studies, exposure to PM contributed to the increased expression of Substance P in the conjunctival tissue, indicating its potential harmful effect on immunological changes [44].

From our findings, the Substance P expression was notably higher in the OVA + PM exposed group followed by the OVA group. Interestingly, this expression was significantly lower in the naive group (Figure 3B,D).

### 3.8. Effect of PM Exposure on Allergic Inflammatory Cytokines in Conjunctiva and Cornea in an AED Mouse Model

The immunofluorescence staining of TNF-α, IL-4 and IL-5 in the conjunctival tissue and TNF-α and IL-4 in the corneal tissue section of the AED mouse model was carried out to show the allergic inflammatory cytokines induced by exposure to PM.

Our study revealed an upregulation in the expression of TNF-α, IL-4, and IL-5 in the OVA + PM exposed group, contrasting significantly with lower expressions observed in the naive group in the conjunctival tissue. Additionally, the expressions in the OVA group closely resembled those in the OVA + PM exposed group (Figure 4A–F). Previous studies align with our findings, demonstrating that the topical administration of PM on the ocular surface increases the expression of TNF-α, IL-4, and IL-5 in ocular allergy models [28,33].

A similar pattern of expression for allergic inflammatory cytokines (TNF-α and IL-4) was observed in the corneal tissue, indicating a parallel impact on ocular health due to the detrimental effects of PM (Appendix A).

Furthermore, a Western blot analysis of p NF-κB was conducted to quantify the protein levels associated with inflammatory responses from the conjunctival tissue in our AED mouse model. Our results revealed heightened protein expression in the OVA + PM exposed groups compared to the naive group. Likewise, the protein expression in the OVA group closely mirrored those observed in the OVA + PM exposed group (Figure 5E,F). Similar findings with an upregulation of allergic inflammatory cytokine levels in the PM-exposed group were observed in the mouse allergic model [3,37,45].

### 3.9. Effect of PM Exposure on Elevated Cell Proliferation in Conjunctiva and Cornea in an AED Mouse Model

The immunofluorescence staining targeting Ki67, a marker for uncontrolled cell proliferation indicative of actively dividing cells, was employed to assess conjunctival and corneal tissue in our AED mouse model. Our investigation revealed a dramatically increased number of Ki67-positive cells in both OVA + PM exposed and OVA groups. Interestingly, this count remained relatively minimal in the naive group. Ki67-positive cells predominantly localized in the basal cell layer of the conjunctival epithelium (Figure 5A,C) and corneal epithelium (Appendix A).

### 3.10. Effect of PM Exposure on Cellular Apoptosis in Conjunctiva and Cornea in an AED Mouse Model

TUNEL staining was employed to assess cellular apoptosis in mouse conjunctival and corneal tissue. Previous studies have highlighted the involvement of PM in cellular apoptosis, both in vitro and in vivo [27]. The elevated presence of apoptotic cells underscores the established harmful effect of PM, accentuating its role in exacerbating allergic responses [27,28].

Our findings showed a comparatively higher number of apoptotic cells in the OVA + PM exposed group followed by OVA group. Intriguingly, the naive group exhibited a significantly diminished count of apoptotic cells in the apical surface of the conjunctival epithelium (Figure 5B,D) and corneal epithelium (Appendix A). 

## 4. Discussion

Numerous studies have shown the detrimental effects of PM on human health. Apart from exacerbating cardiovascular diseases, PM also significantly contributes to respiratory illnesses, thereby amplifying the risk of chronic respiratory morbidity and mortality. [46,47]. Several epidemiological studies and controlled human exposure clinical studies have reported that exposure to PM is harmful to the ocular surface and causes presbyopia, the instability of tear film, AC, toxicity, oxidative stress, and inflammation [27,48,49,50,51]. These symptoms such as dry eye have also been reported in animal studies when applying various types of PM to the eye [1,4]. Thus, this study was designed to elucidate the impacts of PM in air pollution on the ocular surface, considering it as one of the primary organs directly exposed to the environment. With this aim, in the current study, we exposed PM to the eyes of C57BL/6 mice to explore the harmful effects of PM on the ocular surface. Further, this model accurately mirrored the human symptoms both clinically and histopathologically as evidenced by eye rubbing, eyelid and conjunctival edema, ocular secretions, and conjunctival congestion, resembling human allergic reactions. The primary findings of our study are as follows: (1) the OVA-sensitized with PM exposure elevated the production of allergic inflammatory cytokines, (2) rapid uncontrolled cell proliferation, (3) increased mast cell degranulation, (4) elicited vasodilation, (5) elevated production of OVA-specific IgE serum, (6) decreased goblet cell numbers and (7) the upregulation of PMN infiltration in AED mouse model. Thus, our evidence suggests the potential of the OVA-sensitized with PM exposure in the pathogenesis of AED.

Our findings were consistent with those of previous studies on the respiratory system. The previous study on the respiratory allergy mouse model, the combined exposure to allergens (ovalbumin) and PM resulted in a potentiation of allergic responses [52]. Previous studies demonstrated that ambient PM contributed to respiratory inflammation with the increased production of inflammatory mediators and inflammatory cell recruitment [17,53]. The exposure to PM is associated with asthma and allergic respiratory symptoms, but little is known about the influence of PM on ocular allergies [49]. Previously, Lee et al. demonstrated that the topical exposure of 0.5 mg/mL traffic related PM to mice for 14 consecutive days induced symptoms similar to clinical AC, with enhanced mast cells infiltration in the conjunctiva [33]. Similarly, another study indicated an elevation in mast cell numbers, the upregulation of eosinophil infiltration, increased levels of OVA-specific IgE in serum, and heightened clinical scores of allergic conjunctivitis signs. Fujishima et al. noted that PM upregulated an adhesion molecule, chemokines, cytokines (IL-6, IL-8, IL1β), and growth factors in primary cultured healthy human conjunctival epithelium [54]. These findings collectively suggest that exposure to PM exacerbates allergic responses [9].

In our investigation, OVA-sensitized with PM exposure exhibited a stimulatory effect on the release of IgE into peripheral blood, highlighting its potential as a modulator of systemic allergic responses. Likewise, PM exerted a significant influence on goblet cells and Substance P. The observed alterations in goblet cell activity point to a potential disruption in mucosal defense mechanisms, accentuating the vulnerability of the ocular surface [3,27]. Concurrently, the changes in Substance P release highlight a modulation of neuro-immune responses, inflammation, and vascular permeability [31]. These findings suggest that PM exposure can intricately impact both mucosal and neural components, contributing to the complexity of AED [44]. In our study, we observed that ocular allergy manifested as an IgE-mediated immediate hypersensitivity reaction. Our findings align with existing research, highlighting PMN cell inflammation as a central element in the allergic conjunctival response [9]. Our data substantiate the role of conjunctival mast cells as primary mediators, further supporting the connection between these cells and the eosinophilic responses observed in ocular allergies.

Furthermore, our investigation delved into the impact of PM exposure on NF-κB signaling in AED, revealing the activation of this pathway and highlighting its central role in inflammation [55]. Several studies have already proposed a crucial involvement of NF-κB in the development of diseases induced by PM exposure [4,56]. Consistent with prior studies, we demonstrated increased pro-inflammatory mediators like IL-4, IL-5, and TNF-α in our AED mouse model, which were produced by the degranulation of mast cells and emphasized their roles in ocular allergy [57]. Studies have shown that IL-4 and TNF-α promote and maintain the allergic response thereby inducing the apoptosis of the goblet cells [4]. Prominently, IL-4 increased the production of IgE in B-cells and promoted T-cell differentiation into Th2 cells [58,59]. TNF-α promotes the chemotaxis of eosinophils and neutrophils, increasing cytotoxicity and promoting the release of mast cell mediators [58]. In our study, we revealed the intricate connection between the decrease in goblet cells and inflammation in the context of ocular allergy. Our findings emphasize that PM induced irritation and creates a pro-inflammatory environment on the ocular surface. This dynamic interplay underscores the significant role of inflammation in the observed decline in goblet cell numbers, providing crucial insights into the complex interactions within ocular allergy as illuminated by our study.

Additionally, we observed apoptosis primarily in the corneal and conjunctival epithelia of the PM-exposed group, suggesting a link to inflammation, allergic responses, and cellular damage. This phenomenon may involve oxidative stress, inflammation, or direct cellular damage, highlighting a potential association between PM exposure and the onset of ocular allergies [37]. In previous investigations, the presence of ki67 in the model of DED with PM exposure has been well documented [4]. Our study further extends these findings by detecting significantly increased Ki67 expression, indicative of rapid uncontrolled cell proliferation, in the basal epithelia of the cornea and conjunctiva following PM exposure. This new evidence suggests a potential connection between elevated Ki67 levels and the observed heightened cellular proliferation in response to inflammation and amplified allergic responses in ocular tissues. These findings could be attributed to the heightened uptake of PM by ocular macrophages, inducing cellular oxidative stress. Concurrent exposure to OVA and PM during sensitization may potentiate macrophage activation, leading to the increased secretion of pro-inflammatory mediators that regulate early acute inflammatory responses, thereby exacerbating inflammation upon allergen challenge.

Moreover, the mechanisms underlying air pollutant and PM effects on AED, including AC, are not fully understood [60]. Pathologically, the elevated levels of serum IgE antibody and the infiltration of PMN including eosinophils into the conjunctiva are the main pathologic changes in the AED including AC reaction [21]. In the early phase, IgE binds to conjunctival mast cells, triggering an immediate allergic response with the release of IL-4, IL-5, and IL-13 [61]. The late phase is characterized by eosinophil and T-lymphocyte recruitment and activation. Th2-type cells, producing IgE-enhancing cytokines like IL-4 and IL-5, mediate the increased IgE and eosinophil levels in AED mice [62].

Nevertheless, several modifications that can be performed to improve the current OVA-sensitized with PM exposure mouse AED model [41]. Initially, aerosol samples, such as ozone, have been utilized within a gas chamber to provoke an inflammatory reaction on the ocular surface [41]. Similarly, aerosolized PM could be employed within a gas chamber to simulate natural environmental conditions more accurately. Additionally, it is important to consider that environmental exposure to PM differs from topical administration. Therefore, adjustments to concentration and exposure methods should be considered for future studies. Similarly, scratching behavior can be monitored to observe the severity of the allergic responses in the ocular surfaces [3]. Furthermore, a tear measurement was not performed in our study. The brief duration of the experimental period in this study is inadequate. Future studies should extend the duration of the experiments to encompass a longer period to fully capture the long-term effects of PM in inducing ocular allergy. Moreover, this study did not ascertain the specific constituents of PM responsible for exacerbating ocular allergy. Further investigations are needed to identify the different types of chemical components of PM that might exacerbate ocular allergy. Furthermore, differences may exist between the human ocular surface and an animal model. Consequently, a comprehensive, large-scale, long-term clinical study is needed to assess the actual effects of fine ambient PM exposure on the ocular surface. Also, further study is mandatory to investigate the possible mechanism through which PM mediates its immune-toxicological effects. Evidently, future translational research involving ambient PM-exposed human subjects with AED could benefit from considering adjunctive treatments like amniotic membrane therapy [63].

Clinical investigations into the impact of airborne PM on the ocular surface have predominantly focused on discerning detrimental effects. The primary observations entail adverse consequences on the ocular surface, marked by an escalated production of pro-inflammatory cytokines and alterations in tear osmolarity. Consequently, all prior reports regarding PM effects on the ocular surface implicated only dry eye conditions, without addressing the AED pathogenesis [27]. Furthermore, there has been no report on whether PM may induce AED through its natural route, that is, direct contact with the ocular surface, and no report of an AED animal model induced using OVA-sensitized PM (mean diameter < 4 μm) has been published yet. To our knowledge, this is the pioneer AED animal model induced using OVA-sensitized with PM (mean diameter < 4 μm) exposure through direct PM application on the ocular surface.

Despite that, our findings are valuable in terms of elucidating the allergic events underlying PM-induced changes associated with inflammation and allergic responses on the ocular surface and laying the foundation for further translational research involving ambient PM-exposed human subjects with AED. 

## 5. Conclusions

In conclusion, our study successfully utilized OVA with PM exposure to establish an AED model in mice. The findings highlight the potential of PM as a key contributor to AED, shedding light on the intricate relationship between environmental factors and human ocular health. The established murine model, a pivotal outcome of our study, not only confirms the hazardous potential of PM in allergic ocular diseases but also serves as a potent tool for future investigations on different allergic diseases, including asthma and atopic dermatitis. Therefore, increasing PM in the atmosphere will lead to more prevalence of ocular allergy in the future, demanding more research.

## Figures and Tables

**Figure 1 bioengineering-11-00498-f001:**
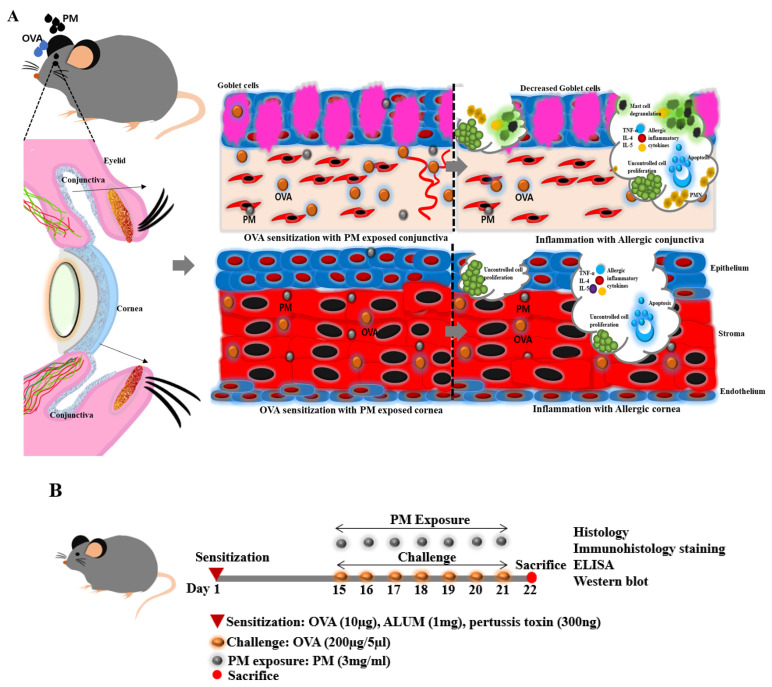
Representation of the co-exposure to PM along with OVA generating immune responses, inflammation, and ocular surface response to PM exposure in a mouse model of AED. (**A**) Schematic illustration of the mouse AED model with OVA sensitized PM exposed conjunctiva and cornea. Inflammation in the conjunctiva and cornea, indicative of allergic responses, was observed following sensitization with OVA and exposure to PM. (**B**) Schematic illustration of in vivo experimental plan. (**C**) Eye features of naive, OVA and OVA + PM exposed groups were assessed by slit-lamp microscopy. (**D**) Changes in the serum IgE concentration level in naive, OVA and OVA + PM exposed groups. In D, data are presented in bar graph with mean ± SEM (*n* = 8, * *p* < 0.05, *** *p* < 0.001), significant difference using one-way ANOVA followed by the Tukey test).

**Figure 2 bioengineering-11-00498-f002:**
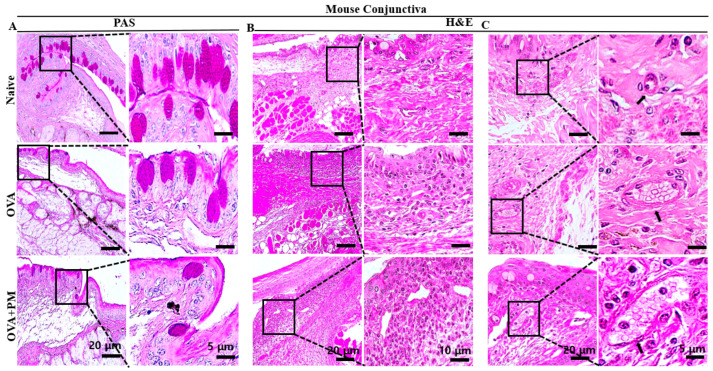
Exposure of PM effects on PMN cell infiltration and vascular dilation in conjunctiva in an AED mouse model. (**A**) PAS staining of conjunctiva performed after AED induction to demonstrate the presence and distribution of the goblet cell in naive, OVA and OVA + PM exposed groups. (**B**) H&E staining of conjunctiva performed after AED induction to show the PMN cell infiltration in naive, OVA and OVA + PM exposed groups. (**C**) H&E staining of conjunctiva performed after AED induction to show the vascular changes in naive, OVA and OVA + PM exposed groups. (**D**) Changes in the goblet cell numbers in the conjunctiva. (**E**) Changes in the PMN cells infiltration in the conjunctiva. (**F**) Changes in the vascular caliber in the conjunctiva. Black rectangles indicate the goblet cells, PMN and vascular dilation area shown in higher-power fields. The black arrowheads indicate the dilated vessels in higher magnification. The scale bar is indicated by a black bar in the lower right corner. In (**D**–**F**), data are presented in bar graph with mean ± SEM (*n* = 5, * *p* < 0.05, ** *p* < 0.01, *** *p* < 0.001), significant difference using one-way ANOVA followed by the Tukey test).

**Figure 3 bioengineering-11-00498-f003:**
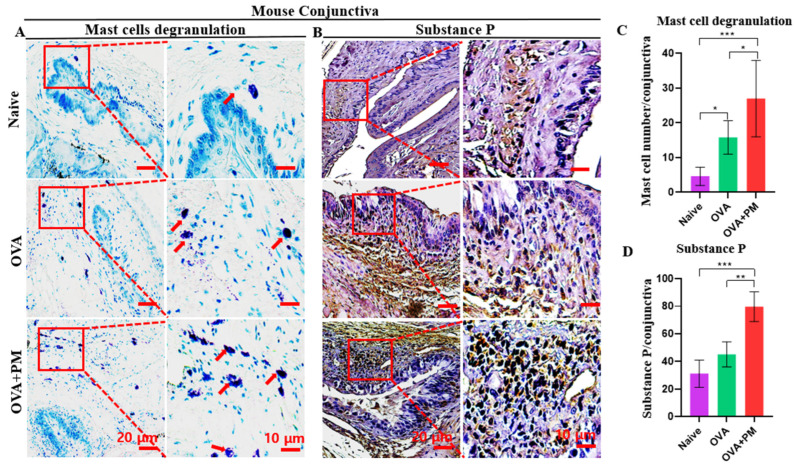
PM effects on mast cell degranulation and Substance P upregulation in an AED mouse model. (**A**) Toluidine blue staining performed after AED induction to demonstrate the conjunctival mast cells in naive, OVA and OVA + PM exposed groups. (**B**) Immunohistochemistry performed after AED induction to demonstrate the Substance P in the conjunctiva in naive, OVA and OVA + PM exposed groups. (**C**) Changes in the number of mast cells degranulation in the conjunctiva. (**D**) Changes in the Substance P positive area in the conjunctiva. Red rectangles indicate mast cell degranulation and Substance P area shown in higher-power fields. The red arrowheads indicate the degranulated mast cells in higher magnification. The scale bar is indicated by a red bar in the lower right corner. In (**C**,**D**) data are presented with bar graph with mean ± SEM (*n* = 4, * *p* < 0.05, ** *p* < 0.01, *** *p* < 0.001, significant difference using one-way ANOVA followed by the Tukey test).

**Figure 4 bioengineering-11-00498-f004:**
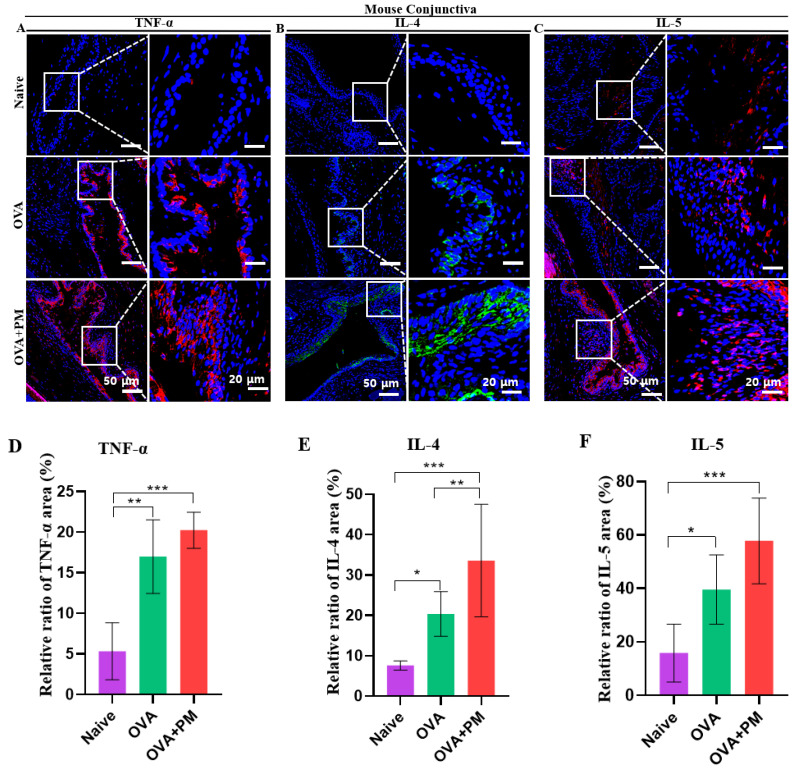
PM exposure triggers allergic anti-inflammatory cytokines infiltration in an AED mouse model. (**A**) Immunofluorescence staining for TNF-α in conjunctiva in naive, OVA and OVA + PM exposed groups. (**B**) Immunofluorescence staining for IL-4 in conjunctiva in naive, OVA and OVA + PM exposed groups. (**C**) Immunofluorescence staining for IL-5 in conjunctiva in naive, OVA and OVA + PM exposed groups. (**D**) Changes in the positive area of TNF- α in conjunctiva. (**E**) Changes in the positive area of IL-4 in conjunctiva. (**F**) Changes in the positive area of IL-5 in conjunctiva. White rectangles indicate TNF-α, IL-4 and IL-5 in conjunctiva; shown in higher-power fields. The scale bar is indicated by a white bar in the lower right corner. In (**D**–**F**), data are presented with bar graph with mean ± SEM (*n* = 5, * *p* < 0.05, ** *p* < 0.01, *** *p* < 0.001, significant difference using one-way ANOVA followed by the Tukey test).

**Figure 5 bioengineering-11-00498-f005:**
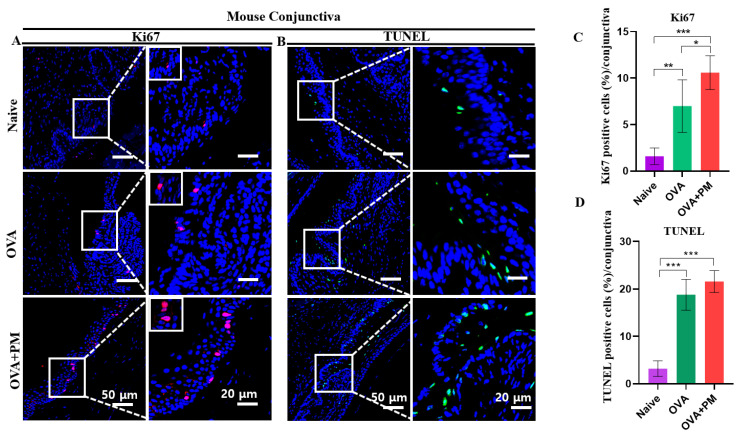
Exposure of PM effects on elevated cell proliferation and cellular apoptosis in an AED mouse model. (**A**) Immunofluorescence staining for ki67 in conjunctiva in naive, OVA and OVA + PM exposed groups. (**B**) TUNEL staining for cellular apoptosis in conjunctiva in naive, OVA and OVA + PM exposed groups. (**C**) Changes in the ki67 positive cells percentage in conjunctiva. (**D**) Changes in the TUNEL positive cells in conjunctiva. (**E**) Western blot showing the expression of p NF-kB in the conjunctiva in naive, OVA and OVA + PM exposed groups. (**F**) Relative expression ratio of p NF-kB to GAPDH. White rectangles indicate the ki67 and TUNEL positive cells in conjunctiva; shown in higher-power fields. The scale bar is indicated by a white bar in the lower right corner. The image located on the upper left in (**A**) is an enlarged version of 20 µm. In (**C**,**D**,**F**) data are presented with bar graph with mean ± SEM (*n* = 5, *n* = 3, * *p* < 0.05, ** *p* < 0.01, *** *p* < 0.001, significant difference using one-way ANOVA followed by the Tukey test). Original images of (**E**) can be found in Appendix A.

**Table 1 bioengineering-11-00498-t001:** Changes in the eye clinical scores of mice in naive, OVA and OVA + PM exposed groups. Results are expressed as mean and standard deviation (*n* = 10).

Score Symptoms	Naive	OVA	OVA + PM
Conjuctival chemosis	0.05 (0.15)	1.33 (1.00)	2.31 (1.10)
Conjuctival hyperemia	0.08 (0.17)	1.67 (0.71)	2.54 (1.02)
Eyelid edema	0.10 (0.14)	1.72 (0.75)	2.67 (1.41)
Tearing and discharge	0.04 (0.06)	1.54 (0.99)	2.56 (1.23)

## Data Availability

All datasets used and/or analyzed during the current study are available from the corresponding author on reasonable request.

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
