# Peer review of "Effect of Exposure to Particulate Matter on the Ocular Surface in an Experimental Allergic Eye Disease Mouse Model"

_bioengineering, 2024, doi:10.3390/bioengineering11050498_

Round 1
Reviewer 1 Report
Comments and Suggestions for Authors
The abstract made more sense than the rest of the manuscript.
The introduction is too long, and does not adequately set up the study. The first paragraph sets up the topic, defining OVA, PM and AED. The second paragraph BRIEFLY cites the relevant publications that have examined these three variables, . The third paragraph states the proposal, the purpose of the study and how it adds to the known literature. This is currently missing, citing a vague figure which make no sense in the context given.
Materials: Too long and uneccessarly detailed.
Clinical scoring can be tabulated (as it can be in the results)
2.2 make no sense: there are three groups: line 151 says "naive group not challenged" then line 153 implies that they were challenged? Figure 1B does not help UNLESS the authors expand it out to show all three conditions (Naive, OVA and OVA+PM).
2.5, 2.6, 2.7 and 2.8 all start the same way. Thus, there need to be a common starting paragraph stating: Animal were sacrificed (how??), eyeballs (how many - all or just one side?) were dissected out and were processed for routine paraffin histology. 5 micron sections were collected using a rotary microtome (brand) and the following stains were applied (to how many sections per animal): H&E (which one?), Toluidine Blue (reference), and PAS (reference). Additional details would be added after this.
For the immuno section - how many sections per animal were stud stained?
TUNEL assay - I presume these were the other half of the eyeballs? How did they get into wax? Were they also prepared for routine paraffin histology?
Western blotting - was this obtained prior to euthanasia, after euthanasia but prior to dissection, or post dissection?
2.11 - test for normality?
Results - there are three experimental conditions, there are three experimental conditions in the figures, but much of the written portion of the results focusses only on OVA+PM (3.1 only describes that group).
Please tabulate clinical scores for all three conditions - easier to read than impenetrable prose.
Please remove excess chatty words like "Next" and "Thus" at starting sections. Also, please refrain from interpreting results (e.g.: lines 408-412, 436-442, 459-461 etc...)
Discussion - the first paragraph is excellent. It clearly lists the 7 major findings of this study. The rest of the discussion is a meandering essay on AED and PM with only occasional reference to the study as presented.
Rewrite the entire discussion ( probably moving paragraphs around) to clearly discuss the significance, and future outlooks of each of the seven findings . Then, end on a conclusion leading to further research in the field.
Figures - tiny and too many. please consolidate into fewer figures, there is no need to show each individual condition for each stain unless there is something really pertinent there.
Author Response
Dear Editor-in-chief,
We really appreciate sending the sincere feedback quickly and the comments that reviewers found interesting to our manuscript “Effects of Exposure to Particulate Matter on the Ocular Surface in an Experimental Allergic Eye Disease Mouse Model (Bioengineering 2975236). We have made corrections focusing on the comments from reviewers. In addition to this, we have checked the entire contents of the manuscript and corrected the sentences. Many edits have been made to reduce the ambiguity of the overall content flow and make it easier for readers to understand. A list of changes is highlighted in red in the revised manuscript.
Response to Reviewer:
Reviewer 1
Reviewer 1 comments
#1. The abstract made more sense than the rest of the manuscript.
We appreciate your time and effort in evaluating our work. We are grateful for your positive assessment of the abstract and understand your concern regarding the clarity of the abstract.
#2. The introduction is too long and does not adequately set up the study. The first paragraph sets up the topic, defining OVA, PM and AED. The second paragraph BRIEFLY cites the relevant publications that have examined these three variables,. The third paragraph states the proposal, the purpose of the study and how it adds to the known literature. This is currently missing, citing a vague figure which makes no sense in the context given.
We are grateful for your insightful comments regarding the structure and clarity of the introduction. We acknowledge your observation that the introduction is too lengthy and lacks sufficient clarity in setting up the study. Upon careful consideration of your feedback, we have revised the introduction to address these concerns in our manuscript.
We have restructured the introduction to ensure brevity while maintaining clarity. We have omitted many sentences and tried to make it shorter.
The first paragraph now succinctly defines the key variables—OVA, PM, and AED—setting up the topic efficiently. (Edit line 45-67)
The small two second and third paragraphs now briefly cite relevant publications that have examined these three variables, providing essential context. (Edit line 68-97)
We recognize that the last paragraph of the introduction section previously lacked a clear statement of the study proposal, its purpose, and its contribution to the existing literature. We revised this section to explicitly outline our research proposal and the aim as well. (Edit line 107-111).
#3. Materials: Too long and unnecessarily detailed.
We appreciate for your valuable feedback on the materials section.
We have revised it accordingly to address your concerns regarding its length and level of detail. Specifically, we have condensed the section by eliminating unnecessary details and emphasizing essential information pertinent to the study.
Additionally, we have shortened lengthy procedures in (materials and method section) to ensure clarity and conciseness while maintaining scientific rigor.
#4. Clinical scoring can be tabulated (as it can be in the results)
We are grateful for your valuable feedback on our manuscript. We appreciate your suggestion regarding the tabulation of clinical scoring in the results section.
We agree that tabulating clinical scoring can enhance the clarity and organization of our findings. Therefore, we included tables to present this information effectively, ensuring that readers can easily interpret and compare the clinical scores. (Table 1, line 305-309, data are presented as mean and standard deviation).
Additionally, a bar graph for the clinical sores from figure 1 is omitted.
#5 2.2 make no sense: there are three groups: line 151 says "naive group not challenged" then line 153 implies that they were challenged? Figure 1B does not help UNLESS the authors expand it out to show all three conditions (Naive, OVA and OVA+PM).
We appreciate for your thorough review and for bringing to our attention the confusion in section 2.2 of the manuscript.
We apologize for the lack of clarity in our description of the experimental groups. We understand how the contradictory statements in lines 151 and 153 may have caused confusion. To address this issue, we revised the text to provide a clear and consistent explanation of the three experimental groups: Naive, OVA, and OVA+PM. (Section 2.2 edit line 127-138)
Additionally, we acknowledge your suggestion to expand Figure 1B to include all three conditions (Naive, OVA, and OVA+PM). We expanded the figure 1B in detail. We believe that this visual representation will enhance the understanding of the experimental setup and facilitate comparisons between the groups.
#6. 2.5, 2.6, 2.7 and 2.8 all start the same way. Thus, there need to be a common starting paragraph stating: Animal were sacrificed (how??), eyeballs (how many - all or just one side?) were dissected out and were processed for routine paraffin histology. 5 micron sections were collected using a rotary microtome (brand) and the following stains were applied (to how many sections per animal): H&E (which one?), Toluidine Blue (reference), and PAS (reference). Additional details would be added after this.
We thank you for your detailed review of our manuscript and for highlighting the need for clarity and consistency in sections 2.5, 2.6, 2.7, and 2.8. We acknowledge your suggestion to include a common starting paragraph to provide clear and consistent information about the experimental procedures. In our revised manuscript, we have started the material and methods sections (2.5, 2.6. 2.7 and 2.8) in a similar way.
We have mentioned similar sentence in our modified manuscript according to the experimental methods applied :
The animals were euthanized by CO2 asphyxiation and left whole eyes were harvested for studying the mast cell degranulation in the mouse conjunctiva. Tissues were then processed for paraffin embedding and sectioning (5 µm) using microtome (Leica, Wetzlar, Germany).
Regarding the section of the tissue used per animals, we randomly selected the 5 samples, and 4 sections were randomly selected from each section (Edit line 212-213). This information is mentioned in the materials and methods section. The number of animals that were used in each experiment is mentioned in the all-figure legends as well.
We have added the manufacturer detailed information for H&E staining in the materials and method section, subsection-2.5(Edit line-174, 176).
We have added the Reference for Toluidine Blue and manufacturer information in our Materials sections, subsection-2.6.
Reference
- Ding, Y.; Li, C.; Zhang, Y.; Ma, P.; Zhao, T.; Che, D.; Cao, J.; Wang, J.; Liu, R.; Zhang, T., Quercetin as a Lyn kinase inhibitor inhibits IgE-mediated allergic conjunctivitis. Food and Chemical Toxicology 2020, 135, 110924.
We have added the Reference for PAS stain along with the manufacturer information in our Materials and methods sections, subsection-2.8(Edit line-224,225)
Reference
- Tan, G.; Li, J.; Yang, Q.; Wu, A.; Qu, D.-Y.; Wang, Y.; Ye, L.; Bao, J.; Shao, Y., Air pollutant particulate matter 2.5 induces dry eye syndrome in mice. Scientific Reports 2018, 8, (1), 17828.
Further we have added further information regarding the Toluidine blue and PAS staining in our methods and materials section.
#7. For the immuno section - how many sections per animal were stud stained?
We appreciate for your insightful feedback on our manuscript.
Regarding the immuno section, we apologize for the oversight in not specifying the number of sections per animal that were stained. We ensured this information in the revised manuscript. We randomly selected the 5 samples, and 4 sections were randomly selected from each section. This information is mentioned in the materials and methods section 2.7, (edit line 212-213).
Typically, in our experimental protocol, we stain a standardized number of sections per animal to maintain consistency and reliability of the results.
#8. TUNEL assay - I presume these were the other half of the eyeballs? How did they get into wax? Were they also prepared for routine paraffin histology?
We are grateful for your valuable feedback and for raising important questions regarding the TUNEL assay methodology in our manuscript.
For the TUNEL assay, tissues were processed for paraffin embedding and sectioning using microtome. They were also prepared by routine paraffin procedures like other procedures (histology, immuno-staining). This information is mentioned in the revised manuscript as stated by the reviewer.
#9. Western blotting - was this obtained prior to euthanasia, after euthanasia but prior to dissection, or post dissection?
We appreciate for your inquiry regarding the timing of conjunctiva sample collection for Western blotting.
To clarify, conjunctiva samples for Western blot analysis were obtained after the euthanasia of the animals. Following euthanasia, conjunctiva samples were collected and processed for Western blot analysis according to the methodology outlined in the Materials and Methods section, subsection 2.9.
#10. 2.11 - test for normality?
We are grateful for your inquiry regarding the test for normality conducted in section 2.11 of our manuscript.
We utilized the Bartlett test to assess the effects of multiple treatments in in vivo experiments, with p-values < 0.05 were considered statistically significant. This information is mentioned in the revised manuscript subsection 2.11, (Edit line 265-266)
#11. Results - there are three experimental conditions, there are three experimental conditions in the figures, but much of the written portion of the results focusses only on OVA+PM (3.1 only describes that group).
We appreciate for your insightful feedback on our manuscript. We acknowledge your observation regarding the discrepancy between the three experimental conditions presented in the figures and the focus on only one condition, OVA+PM, in much of the written portion of the results, particularly in section 3.1.
To address this concern, we have revised all the results section to ensure adequate coverage of all three experimental conditions—Naive, OVA, and OVA+PM. Specifically, we expanded section 3.1 and all other remaining sections in our revised manuscript to provide detailed descriptions of the results for each experimental condition, ensuring equitable attention to all groups.
Section 3
Section 3.1 (Edit lines -278-288)
Section 3.2 (Edit lines-318-320)
Section 3.3 (Edit lines-329-331)
Section 3.4 (Edit lines-335-340)
Section 3.5 (Edit lines-361-365)
Section 3.6 (Edit lines-373-375)
Section 3.7 (Edit lines-384-386)
Section 3.8 (Edit lines-405-408,413-415, 419-421)
Section 3.9 (Edit lines-428-432)
Section 3.10 (Edit lines-458-462)
#12. Please tabulate clinical scores for all three conditions - easier to read than impenetrable prose.
We thank you for your suggestion to tabulate the clinical scores for all three experimental conditions in the Results section.
We agree that tabulating the clinical scores will enhance readability and facilitate a clearer understanding of the results. Therefore, we included a table in the Results section (Table 1, line 305-309,) that presents the clinical scores for each experimental condition—Naive, OVA, and OVA+PM. The data are presented as Mean and standard deviation.
#13. Please remove excess chatty words like "Next" and "Thus" at starting sections. Also, please refrain from interpreting results (e.g.: lines 408-412, 436-442, 459-461 etc...)
We thank you for your feedback and for highlighting areas where excess chatty words and interpretation of results are present.
We carefully reviewed the text to remove unnecessary words like "Next" and "Thus" from the starting sections, as well as citated the sentences from the already published articles.
We further refrain from interpreting results in the manner indicated in all result sections and omitted the sentences in our revised manuscript as stated by the reviewer.
#14. Discussion - the first paragraph is excellent. It clearly lists the 7 major findings of this study. The rest of the discussion is a meandering essay on AED and PM with only occasional reference to the study as presented.
We are grateful for your positive feedback on the first paragraph of the Discussion section, specifically noting its clarity in listing the major findings of our study.
We appreciate your constructive criticism regarding the subsequent content of the Discussion.
To address this, we revised the Discussion section to ensure a more focused and concise analysis of our study's results. We have deleted many unnecessary sentences from our revised manuscript.
Further, we maintained a clear connection between our findings and the broader context of AED and PM, while also emphasizing the relevance of our results to the field.
#15. Rewrite the entire discussion (probably moving paragraphs around) to clearly discuss the significance, and future outlooks of each of the seven findings. Then, end on a conclusion leading to further research in the field.
We greatly appreciate for your insightful feedback on our Discussion section. We have carefully considered your suggestions and have rewritten and moved the paragraphs and omitted many unnecessary sentences to provide a clearer and more focused analysis of each of the seven major findings of our study. The edited paragraphs are highlighted with red color.
We have also discussed in terms of its significance and potential implications for future research in our conclusion (Edit line 604-607, 612-616)
#16. Figures - tiny and too many. please consolidate into fewer figures, there is no need to show each individual condition for each stain unless there is something pertinent there.
We appreciate your suggestion to consolidate the figures to reduce their number and improve readability.
In response, we have consolidated into fewer figures, while also moving some of the less pertinent images to the supplementary materials. This approach will ensure that the main figures focus on presenting the most relevant and impactful data, while additional images are still available for reference in the supplementary materials (Supplementary figure S1-as indicated in line number -415, S2- as indicated in line number 432, and 461-462). Additionally, we have employed high magnification in the figures to ensure clear visualization.
I am grateful for the reviewer’s advice.
With best regards,
Jae Yong Kim, MD, PhD
Professor
Department of Ophthalmology, Asan Medical Center,
University of Ulsan College of Medicine, Seoul 05505, Republic of Korea
Email: jykim2311@amc.seoul.kr

Reviewer 2 Report
Comments and Suggestions for Authors
Please, give also other methods:
- amniotic membrane by DED is necessary to be mentioned or dry eye after tumor excision
Comments on the Quality of English Language
Please, give better qaulity pictures by original Western blot images
Author Response
Dear Editor-in-chief,
We really appreciate sending the sincere feedback quickly and the comments that reviewers found interesting to our manuscript “Effects of Exposure to Particulate Matter on the Ocular Surface in an Experimental Allergic Eye Disease Mouse Model (Bioengineering 2975236). We have made corrections focusing on the comments from reviewers. In addition to this, we have checked the entire contents of the manuscript and corrected the sentences. Many edits have been made to reduce the ambiguity of the overall content flow and make it easier for readers to understand. A list of changes is highlighted in red in the revised manuscript.
Response to Reviewer:
Reviewer 2
Reviewer 2 comments
#1. Please, give also other methods:
- amniotic membrane by DED is necessary to be mentioned or dry eye after tumor excision
We appreciate your insightful comments regarding the amniotic membrane by DED or dry eye after tumor excision. We have mentioned this information in our Discussion section (Edit line 590-592)
#2. Please, give better quality pictures by original Western blot images.
We appreciate your valuable feedback on our manuscript. We have inserted the better-quality figure of Western blot in Figure 5E.
I am grateful for the reviewer’s advice.
With best regards,
Jae Yong Kim, MD, PhD
